# Validation of the Romanian version of the brief negative symptom scale in a heterogeneous schizophrenia inpatient sample

Cosmin Ioan Moga[1] ⬤, Denisa Gliția[2], Octavia Oana Căpățînă[1], Cătălina Angela Crișan[1], Mihaela Fadygas-Stănculete[1] and Ioana Valentina Micluția[1]

[1]Iuliu Hațieganu University of Medicine and Pharmacy of Cluj-Napoca, Romania and [2]Clinic I of Psychiatry of Cluj-Napoca, Romania

## Research Article

BNSS; cross-cultural; negative symptoms; Romanian; schizophrenia

**Corresponding author:**
Cosmin Ioan Moga;
Email: moga_cosmin_33@yahoo.com

## Abstract

Negative symptoms in schizophrenia are critical to functional outcomes but remain difficult to assess reliably. The Brief Negative Symptom Scale (BNSS) was developed to address these challenges, though no validation exists in Romanian-speaking populations. To validate the BNSS in a Romanian clinical sample, explore its psychometric properties and compare BNSS-based and PANSS-based classifications of severe negative symptoms. Forty-seven inpatients with schizophrenia were assessed using Romanian versions of the BNSS, PANSS, CDSS and AIMS. Psychometric analyses included internal consistency, inter-rater reliability, factor analysis and correlation-based validity. Two classification schemes, moderate–severe negative symptoms, measured by BNSS (BNSS-MS), and predominant negative symptoms, measured by PANSS (PANSS-PNS), were compared. The BNSS showed excellent internal consistency ($\alpha = .94$) and inter-rater reliability (ICC = .98). A five-factor structure was confirmed. BNSS total scores correlated strongly with PANSS negative ($\rho = .90$), but not with positive, depressive, or motor symptoms. Blunted affect emerged as the most prominent subscale. The BNSS-MS group captured more severe cases than PANSS-PNS and showed greater symptom burden and higher distress scores. The Romanian BNSS is valid and sensitive for detecting negative symptoms, outperforming PANSS in identifying clinically significant subgroups.

## Impact statement

This study offers the first validation of the Brief Negative Symptom Scale (BNSS) in a Romanian-speaking population and one of the few conducted in a heterogeneous schizophrenia sample, including acutely ill patients. The findings confirm the BNSS as a psychometrically robust tool for assessing negative symptoms across all phases of illness. Comparative analyses demonstrate its enhanced sensitivity over the PANSS in detecting clinically meaningful negative symptomatology, even using minimal severity thresholds. Blunted affect emerged as a particularly prominent and discriminative domain, underscoring its salience among psychotic features. These results support the BNSS as a valuable instrument for identifying deficit phenotypes and informing targeted clinical assessment and intervention strategies.

## Introduction

Rooted in 19th-century psychopathological descriptions and embedded in Kraepelin's original conceptualization of schizophrenia (Kraepelin, 1921), negative symptoms refer to a deficit or absence of conative functions and remain a key predictor of poor prognosis in schizophrenia (Kirkpatrick et al., 2006; Rabinowitz et al., 2013; Mucci et al., 2019). To identify clinically relevant subgroups within schizophrenia characterized by negative symptoms, various classifications have been proposed based on etiological and severity-related assumptions. These include nosographic frameworks, such as Crow's type II *versus* type I schizophrenia (Crow, 1985) and Carpenter's deficit *versus* non-deficit schizophrenia (Carpenter et al., 1988). Other models are primarily psychometric, including constructs like persistent negative symptoms, predominant negative symptoms, or prominent negative symptoms (Bucci and Galderisi, 2017; Marder and and Galderisi, 2017). Etiologically oriented distinctions have also been advanced, particularly the differentiation between primary negative symptoms – those intrinsic to the illness – and secondary negative symptoms, which arise from factors such as depression, positive symptoms, or medication side effects (Brian Kirkpatrick, 2014).

 The current conceptualization of negative symptoms was refined by the National Institute of Mental Health-Measurement and Treatment Research to Improve Cognition in Schizophrenia

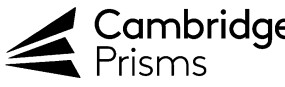



(NIMH-MATRICS) Consensus in 2005, which identified five core domains: anhedonia, asociality, avolition, blunted affect and alogia (Kirkpatrick et al., 2006). Factor analytic studies have consistently supported a two-factor model, grouping anhedonia, avolition and asociality into the motivational deficit domain (MAP) and blunted affect and alogia into the expressive deficit domain (EXP) (Weigel et al., 2023). However, more recent evidence suggests that a hierarchical five-factor model – with MAP and EXP as second-order dimensions overlying the five core domain – may offer a more comprehensive and cross-culturally robust structural representation of negative symptoms (Gehr et al., 2019b).

The BNSS is a second-generation instrument developed to align with the five-domain framework of negative symptoms established by the NIMH-MATRICS Consensus. It was conceived around seven key principles (Kirkpatrick et al., 2011): brevity, full coverage of five core domains, cross-cultural use, applicability beyond trials, differentiation of anhedonia types, separation of experience *vs.* behavior and exclusion of disorganization-related items. The scale also includes a lack of normal distress item (I4), which, despite not loading onto core factors, has clinical value in distinguishing primary from secondary negative symptoms (Kirkpatrick et al., 2011). The BNSS has shown robust psychometric properties, including excellent reliability and strong convergent and discriminant validity with other various established instruments (Kirkpatrick et al., 2011; Strauss et al., 2012; Mucci et al., 2019; Weigel et al., 2023), along with an enhanced sensitivity for identifying the severe negative symptoms groups (Mucci et al., 2019).

Although the scale has been translated into multiple languages (Tatsumi et al., 2020), cross-cultural validation studies remain relatively limited (Weigel et al., 2023), as only a handful of studies have rigorously examined its validity across diverse cultural contexts (Mané et al., 2014; Mucci et al., 2015; Bischof et al., 2016; de Medeiros et al., 2019; Hashimoto et al., 2019; Wójciak et al., 2019; Gehr et al., 2019a; Jeakal et al., 2020; Seelen-De Lang et al., 2020; Métivier et al., 2025). Cross-cultural validation is essential to adequately assess the universality and cultural robustness of the negative symptom construct. To date, no published validation of the BNSS has been conducted within a Romanian population.

This study had three primary objectives. First, we aimed to validate the BNSS in a Romanian-speaking population. To enhance the generalizability of the scale, we included a clinical sample spanning various levels of illness severity, from stable inpatients to individuals in a moderately psychotic phase of schizophrenia. Second, we examined the relationship between BNSS-assessed negative symptoms and a range of sociodemographic, clinical and symptom dimensions. Third, we sought to identify a subgroup of patients with pronounced negative symptoms using BNSS-defined moderate-to-severe thresholds and to compare this group with those classified by the PANSS-based predominant negative symptom algorithm.

## Material and methods

### Participants

The study sample consisted of adult inpatients recruited from Psychiatric Clinics I and II in Cluj-Napoca, Romania. **Inclusion criteria** were: (1) a diagnosis of schizophrenia according to the Diagnostic and Statistical Manual of Mental Disorders, Fifth Edition (DSM-5), based on the clinical evaluations and standardized neuropsychiatric interview algorithms used by the admitting units; (2) age between 18 and 65 years; and (3) voluntary hospitalization with the capacity to provide informed consent. **Exclusion criteria**

consisted of: (1) an estimated IQ below 70; (2) a history of neurological disorders; (3) current alcohol or other substance dependence; (4) comorbid severe mental disorders; and (5) insufficient fluency in Romanian.

### Instruments

The study utilized four standardized clinical measures; the **Brief Negative Symptom Scale (BNSS)** (Kirkpatrick et al., 2011) is a 13-item clinician-rated instrument designed to assess negative symptoms across five core domains – anhedonia, asociality, avolition, blunted affect and alogia – alongside an additional item evaluating distress. Each item is rated on a 7-point scale from 0 (absent) to 6 (severe). The **Positive and Negative Syndrome Scale (PANSS)** (Kay et al., 1987) is a 30-item instrument comprising three subscales: positive (PANSS-P), negative (PANSS-N) and general psychopathology (PANSS-G), with respective scoring ranges of 7–49 for PANSS-P and PANSS-N and 16–112 for PANSS-G. PANSS was used to assess overall symptom severity and to establish convergent validity (*via* PANSS-N) and discriminant validity (*via* PANSS-P and PANSS-G). To evaluate depressive symptoms and motor side effects, the **Calgary Depression Scale for Schizophrenia (CDSS)** (Addington et al., 1990) and the **Abnormal Involuntary Movement Scale (AIMS)** (Guy, 1976) were administered, respectively. Both scales were included as control variables for the analysis of discriminant validity.

### Group definitions

**Predominant negative symptoms based on PANSS criteria (PANSS-PNS)** were defined following European Psychiatric Association (EPA) guidelines, requiring either: (1) at least three moderate or two moderately severe negative symptoms, or (2) a PANSS negative score exceeding the positive subscale by $\geq 6$ points, (3) a negative score $\geq 21$ and $\geq 1$ point higher than positive, or (4) any negative score higher than positive. Secondary symptoms were excluded by ensuring PANSS positive $< 19$ and low levels of depression (CDSS $\leq 6$) and motor symptoms (AIMS $< 1$) (Galderisi et al., 2021).

The **moderate–severe negative symptom group (BNSS-MS)** was defined as having BNSS subscale scores $\geq 3$ across all five domains (Mucci et al., 2019).

### Translation and cultural adaptation

The Romanian cultural adaptation and validation of the BNSS (translated in Romanian as "*Scala scurtă a simptomelor negative*") followed the COnsensus-based Standards for the selection of health Measurement INstruments (COSMIN) guidelines (Mokkink, 2018). Permission and a preliminary translation were obtained from the original authors. As a revised Romanian version was already under development by the original team, this version was used with their approval. Additionally, two Romanian-speaking psychiatrists independently translated the workbook and manual, which were then compared against both the original English and the preliminary Romanian versions. A back-translation was performed by bilingual professionals unfamiliar with the BNSS. A multidisciplinary panel, including members of the original translation team and Romanian psychiatrists, reviewed all versions. Minor lexical adjustments were made to enhance clarity and naturalness in Romanian, without altering the essential semantic content.

## Procedure

Two trained psychiatrists conducted single-session assessments using the Romanian-translated versions of the BNSS, PANSS, CDSS, and AIMS. Both raters received formal training in the administration and scoring of the BNSS to ensure consistency. Inter-rater reliability was evaluated through double ratings, which were conducted for a subsample of 10 patients.

## Statistical analyses

All analyses were conducted in R (version 4.4.2). Normality was assessed using the Shapiro–Wilk test to inform the use of non-parametric statistics. Descriptive statistics included means (SD) and frequencies (%). BNSS psychometric properties were evaluated through exploratory (EFA) and confirmatory (CFA) factor analyses, with EFA using minimum residual extraction and oblimin rotation and CFA testing one-, two- and five-factor models. Convergent and discriminant validity were examined using Spearman and partial correlations (controlling for CDSS and AIMS). Internal consistency was assessed using Cronbach's alpha. Inter-rater reliability of the BNSS was assessed using the intraclass correlation coefficient (ICC), calculated based on ratings from a subsample of 10 patients who were evaluated independently by both raters. Group comparisons were conducted using parametric ($t$-test, analysis of variance [ANOVA]) or non-parametric (Mann–Whitney U test/Wilcoxon rank-sum test, Kruskal–Wallis, and Friedman test) methods, depending on normality assumptions.

## Ethical considerations

The study followed the principles outlined in the Declaration of Helsinki. All participants gave informed consent, and the university's ethics committee approved the protocol.

## Results

### Descriptive statistics

Demographic and clinical characteristics appear in Table 1. The 47 inpatients (59.57% female) were on average 41.70 years old (SD = 12.00) and had been ill for 15.06 years (SD = 10.96). Mean PANSS positive, negative and total scores were 21.79 (SD = 6.23), 27.11 (SD = 7.50) and 93.28 (SD = 17.35), respectively; the BNSS total averaged 39.67 (SD = 17.11).

### Scale validation

Sampling adequacy was meritorious, as indicated by a Kaiser–Meyer–Olkin (KMO) value of .86 and a significant Bartlett's test of sphericity ($\chi^2$[78] = 506.83, $p$ < .001). Parallel analysis suggested a dominant general factor; however, exploratory factor analysis (EFA) revealed a more differentiated structure. A one-factor solution produced high item saturation (loadings = .55–.95), but demonstrated poor fit (Tucker–Lewis Index [TLI] = .67, root mean square error of approximation [RMSEA] = .19), and lack of normal distress item showed a notably lower loading (.55) and communality (.30). A two-factor solution aligned with the established MAP and EXP dimensions, explaining 64% of the variance, with moderately correlated factors (r = .65) and improved fit indices (TLI = .80, RMSEA = .15, 90% confidence interval [CI] [.11, .19]). Confirmatory factor analysis (CFA) was used to compare

**Table 1.** Demographic and illness-related characteristics of the study sample (N = 47)

| Variable | Category | n | % | M | SD |
|---|---|---|---|---|---|
| Gender | Female | 28 | 59.57 | | |
| | Male | 19 | 40.43 | | |
| Living environment | Urban | 39 | 82.98 | | |
| | Rural | 8 | 17.02 | | |
| Education level | Secondary | 28 | 59.57 | | |
| | University | 18 | 38.30 | | |
| | Primary | 1 | 2.13 | | |
| Civil status | Single | 36 | 76.60 | | |
| | Married | 11 | 23.40 | | |
| Social status | Unemployed | 40 | 85.11 | | |
| | Employed | 7 | 14.89 | | |
| Age (years) | | | | 41.70 | 12.00 |
| Age of illness onset (years)[a] | | | | 26.64 | 7.49 |
| Illness duration (years)[a] | | | | 15.06 | 10.96 |
| PANSS positive | | | | 21.79 | 6.23 |
| PANSS negative | | | | 27.11 | 7.50 |
| PANSS general | | | | 44.38 | 8.91 |
| PANSS total score | | | | 93.28 | 17.35 |
| BNSS total score[a] | | | | 39.67 | 17.11 |
| BNSS MAP[a] | | | | 21.41 | 8.96 |
| BNSS EXP[a] | | | | 15.65 | 8.74 |
| Anhedonia | | | | 8.73 | 4.39 |
| Lack of normal distress[a] | | | | 2.61 | 1.40 |
| Asociality | | | | 6.46 | 2.92 |
| Avolition | | | | 6.22 | 2.74 |
| Blunted affect[a] | | | | 10.47 | 5.57 |
| Alogia[a] | | | | 5.18 | 3.88 |
| CDSS score[a] | | | | 2.00 | 2.53 |
| AIMS score[a] | | | | 0.13 | 0.49 |

Abbreviations: M = mean; SD = standard deviation; PANSS = Positive and Negative Syndrome Scale; BNSS = Brief Negative Symptom Scale; MAP = motivational factor; EXP = expressive factor; CDSS = Calgary Depression Scale for Schizophrenia; AIMS = Abnormal Involuntary Movement Scale.
[a]Non-normal distribution as showed by Shapiro–Wilk test results ($p$ < .05).

alternative structural models. Among the one-, two- and five-factor models tested, the five-factor solution – reflecting the original theoretical domains of the BNSS – demonstrated the best fit ($\chi^2$ = 66.02, degree of freedom [df] = 44, $p$ = .02, CFI = .95, TLI = .93, RMSEA = .10, 90% CI [.04, .15] standardized root mean square residual [SRMR] = .05), with standardized loadings ranging from 0.77 to 0.99.

The BNSS total score correlated strongly with PANSS negative ($\rho$ = .90, $p$ < .001, 95% [.78, .96]) and moderately with PANSS total ($\rho$ = .55, $p$ < .001, 95% CI [.27, .73]). Correlations with PANSS positive ($\rho$ = −.12, 95% CI [−.39, .15]), CDSS ($\rho$ = .15, 95% CI [−.17, .45]) and AIMS ($\rho$ = .12, 95% CI [−.03, .32]) were small and nonsignificant ($p$ > .30). In contrast, the correlation with PANSS

**Table 2.** Summary of BNSS Validation

| Step | Results |
|---|---|
| KMO & Bartlett's Test | KMO = 0.86; Bartlett's $\chi^2$ (78) = 506.83, $p$ < .01 |
| EFA (2-factor) | MAP: I1–I8, EXP: I9–I13; loadings ≥ .55; variance explained = 64%; RMSEA = .15 (90% CI [.11, .19]) |
| CFA (5-factor) | CFI = .96, TLI = .93, RMSEA = .10 (90% CI [.04, .15]), SRMR = .05 |
| Internal consistency | $\alpha$ = .94 (95% CI [.91, .96]); anhedonia = .91 [.85, .95]; asociality = .82 [.67, .90]; avolition = .80 [.63, .89]; blunted affect = .92 [.87, .95]; alogia = .93 [.87, .96]; ICC(A,1) = .98 (95% CI [.42, 1.00])[a] |
| Convergent validity (BNSS total) | PANSS negative $\rho$ = .90 ($p$ < .001, 95% CI [.78, .96]); PANSS total $\rho$ = .55 ($p$ < .001, 95% CI [.27, .73]) |
| Discriminant validity (BNSS total) | PANSS positive $\rho$ = −.12 ($p$ = .42, 95% CI [−.39, .15]); PANSS general $\rho$ = .42 ($p$ = .02, 95% CI [.10, .65]); CDSS $\rho$ = .15 ($p$ = .31, 95% CI [−.17, .45]); AIMS $\rho$ = .12 ($p$ = .41, 95% CI [−.03, .32]) |
| Partial correlations | BNSS–PANSS negative partial $\rho$ = .92 ($p$ < .01, 95% CI [.86, .96]) controlling for CDSS & AIMS |

Abbreviations: KMO = Kaiser–Meyer–Olkin test for sampling adequacy; $\chi^2$ = Bartlett's test of sphericity; EFA = exploratory factor analysis; MAP = motivation deficit factor; EXP = expressive deficit factor; CFA = confirmatory factor analysis; CFI = comparative fit index; TLI = Tucker–Lewis Index; RMSEA = root mean square error of approximation; SRMR = standardized root mean square residual; $\alpha$ = Cronbach's alpha (internal consistency); ICC = intraclass correlation coefficient; BNSS = Brief Negative Symptom Scale; PANSS = Positive and Negative Syndrome Scale; CDSS = Calgary Depression Scale for Schizophrenia; AIMS = Abnormal Involuntary Movement Scale.
[a]Performed only for a subset of $n$ = 10; $\rho$ = Spearman's rho.

general psychopathology was moderate and statistically significant ($\rho$ = .42, $p$ = .02, 95% CI [.10, .65]). The strong relationship between BNSS and PANSS negative remained robust when controlling for depressive and extrapyramidal symptoms (partial $\rho$ = .92, $p$ < .001, 95% CI [.86, .96]).

Internal consistency was high, with Cronbach's alpha values of .94 (95% CI [.91, .96]) for the BNSS total score and .80–.93 across subscales. Inter-rater reliability, assessed in a subsample of 10 participants with double ratings, yielded an intraclass correlation coefficient of ICC(A,1) = .98, 95% CI [.42, 1.00], indicating excellent agreement between raters. A summary of the validation results is presented in Table 2.

### Sociodemographic and symptom associations

BNSS subscale scores did not differ by gender, education, residence or marital status; unemployed participants scored higher on avolition (mean = 3.29 *vs.* 2.07, t = −2.49, $p$ = .03, 95% CI [−4.66, −0.23]). Age, age at onset and illness duration were unrelated to BNSS domains ($\rho$ ≤ .22). Correlational analyses (Table 3) confirmed strong associations between all BNSS subscales and the PANSS negative dimension ($\rho$ = .68–.86, $p$ < .01), with blunted affect showing the highest correlation. Within the PANSS general scale, several significant item-level correlations were observed: blunted affect with motor retardation ($\rho$ = .50) and disturbance of volition ($\rho$ = .44), anhedonia with disturbance of volition ($\rho$ = .48), asociality with social avoidance ($\rho$ = .57) and alogia with motor retardation ($\rho$ = .51). Blunted affect was rated significantly higher than alogia in the full sample ($p$ < .01) and remained elevated in the BNSS-MS group ($p$ = .01), but not in PANSS-PNS, suggesting reduced subscale differentiation within specific negative symptom definitions (see Figure 1). BNSS subscales showed no significant correlations with CDSS ($\rho$ = .06–.20, $p$ > .15), and only blunted

affect correlated with AIMS ($\rho$ = .29, $p$ = .05), suggesting minimal overlap with extrapyramidal symptoms. Finally, the BNSS lack of normal distress item correlated moderately and significantly with the PANSS negative subscale ($\rho$ = .45, $p$ < .01), but not with PANSS general overall or CDSS scales. Nonetheless, it showed negative correlations with several general psychopathology individual items, including somatic concern ($\rho$ = −.35), anxiety ($\rho$ = −.38), guilt feelings ($\rho$ = −.30), tension ($\rho$ = −.26), and depression ($\rho$ = −.30), along with a negligible association with total CDSS score ($\rho$ = −.04).

### Grouping analyses

Using the BNSS moderate-severity (BNSS-MS) criterion, 15 participants (31.91%) met the threshold for moderate negative symptoms, while the PANSS-defined predominant negative symptom group (PANSS-PNS) included 13 individuals (27.66%). Six participants (12.77%) met both criteria, and 25 (53.19%) met neither, indicating slight overlap (Cohen's κ = 0.19, 95% CI [0.08–0.30]). A McNemar test revealed no directional bias between methods ($\chi^2$ = 0.06, $p$ = .80). Tests of variance homogeneity found no significant differences in BNSS total scores between BNSS-MS and PANSS-PNS groups (Levene F = 2.99, $p$ = .09; Fligner $\chi^2$ = 2.12, $p$ = .15), suggesting comparable dispersion. The two groups were analyzed independently despite partial overlap. As shown in Figure 2, the BNSS-MS group showed significantly higher scores on BNSS total (mean = 56.77, $p$ < .01, 95% CI [4.00, 18.00]), PANSS negative total (mean = 34.00, $p$ = .01, 95% CI [1.68, 10.62]) and PANSS positive total (mean = 20.60, $p$ = .01, 95% CI [1.64, 9.10]) compared to the PNS-PANSS group (BNSS = 43.96; PANSS-N = 27.85; PANSS-P = 15.23). Differences were also observed between MS and non-MS participants on BNSS (56.77 *vs.* 31.66) and PANSS-N (34.0 *vs.* 23.88), both $p$ < .01. In contrast, PNS-PANSS participants did not differ significantly from non-PNS individuals on either BNSS total ($p$ = .32) or PANSS-N ($p$ = .64). Regarding contamination indices (positive, general, depressive, motor symptoms and the BNSS lack of distress item), the BNSS-MS group differed significantly from non-MS participants only on the lack of normal distress item (mean = 3.73 *vs.* 2.08, $p$ < .01, 95% CI [1.00, 2.00]); all other comparisons were nonsignificant ($p$ > .37). The PNS group scored significantly lower than non-PNS participants on positive ($p$ < .01, 95% CI [−11.50, −6.63]) and general subscales ($p$ = .01, 95% CI [−11.92, −1.69]), but did not differ on CDSS ($p$ = .27), AIMS ($p$ = .29), or lack of normal distress ($p$ = .52).

### Discussion
#### Main findings

This study yielded four key findings regarding the assessment and characterization of negative symptoms in schizophrenia. (1) The BNSS demonstrated strong psychometric performance in a Romanian-speaking clinical sample, including a factor structure consistent with theoretical models, excellent convergent and discriminant validity and high internal consistency and inter-rater reliability. (2) Blunted affect emerged as a particularly prominent domain, showing significantly higher mean scores than other BNSS subscales and the strongest correlations with both the PANSS negative and general psychopathology dimensions. (3) Sociodemographic and clinical variables had a limited impact on negative symptoms, with *Avolition* being the only domain significantly associated with employment status, suggesting specific functional relevance. (4) The BNSS-defined moderate-severity group (BNSS-MS)

**Table 3.** Spearman Correlations ($\rho$) Between BNSS Subscales and PANSS, CDSS and AIMS

| PANSS dimensions/ other scales | Anhedonia | Asociality | Avolition | Blunted affect | Alogia | Lack of normal distress |
|---|---|---|---|---|---|---|
| PANSS positive | −0.13, [−0.40, 0.17] (0.40) | 0.01, [−0.28, 0.29] (0.96) | −0.03, [−0.32, 0.26] (0.82) | 0.00, [−0.28, 0.29] (0.98) | −0.23, [−0.49, 0.06] (0.11) | 0.01, [−0.28, 0.29] (0.96) |
| PANSS negative | 0.73, [0.57, 0.84] (<.01) | 0.72, [0.54, 0.83] (<.01) | 0.68, [0.49, 0.81] (<.01) | 0.86, [0.76, 0.92] (<.01) | 0.73, [0.56, 0.84] (<.01) | 0.45, [0.19, 0.65] (<.01) |
| PANSS general | 0.35, [0.07, 0.58] (0.01) | 0.26, [−0.03, 0.51] (0.08) | 0.36, [0.08, 0.59] (0.01) | 0.53, [0.28, 0.71] (<.01) | 0.25, [−0.04, 0.50] (0.09) | −0.01, [−0.30, 0.28] (0.95) |
| Somatic concern | 0.10, [−0.19, 0.38] (0.49) | −0.04, [−0.33, 0.25] (0.77) | −0.00, [−0.29, 0.28] (0.98) | −0.02, [−0.31, 0.27] (0.89) | −0.10, [−0.38, 0.19] (0.51) | −0.35, [−0.58, −0.07] (0.02) |
| Anxiety | 0.12, [−0.18, 0.39] (0.44) | −0.01, [−0.30, 0.28] (0.95) | −0.06, [−0.34, 0.23] (0.70) | 0.01, [−0.28, 0.30] (0.94) | −0.12, [−0.39, 0.18] (0.43) | −0.38, [−0.60, −0.10] (<.01) |
| Guilt | −0.14, [−0.41, 0.15] (0.34) | −0.22, [−0.47, 0.08] (0.15) | −0.08, [−0.36, 0.21] (0.59) | −0.06, [−0.34, 0.23] (0.69) | −0.10, [−0.37, 0.19] (0.51) | −0.30, [−0.54, −0.02] (0.04) |
| Tension | 0.05, [−0.24, 0.33] (0.73) | −0.08, [−0.36, 0.22] (0.61) | 0.06, [−0.23, 0.34] (0.71) | 0.03, [−0.26, 0.31] (0.86) | −0.08, [−0.36, 0.21] (0.58) | −0.26, [−0.51, 0.03] (0.08) |
| Mannerisms | 0.24, [−0.05, 0.49] (0.11) | 0.19, [−0.10, 0.45] (0.21) | 0.41, [0.14, 0.62] (<.01) | 0.39, [0.11, 0.61] (<.01) | 0.13, [−0.16, 0.40] (0.38) | 0.02, [−0.27, 0.31] (0.88) |
| Depression | 0.12, [−0.17, 0.40] (0.40) | −0.08, [−0.36, 0.21] (0.57) | 0.12, [−0.17, 0.39] (0.42) | 0.24, [−0.05, 0.49] (0.11) | 0.04, [−0.25, 0.32] (0.79) | −0.30, [−0.54, −0.02] (0.04) |
| Motor retard | 0.37, [0.09, 0.59] (0.01) | 0.27, [−0.01, 0.52] (0.06) | 0.33, [0.05, 0.56] (0.02) | 0.50, [0.25, 0.69] (<.01) | 0.51, [0.26, 0.70] (<.01) | 0.17, [−0.12, 0.43] (0.26) |
| Uncooperative | 0.30, [0.02, 0.54] (0.04) | 0.13, [−0.17, 0.40] (0.40) | 0.25, [−0.04, 0.50] (0.09) | 0.22, [−0.08, 0.47] (0.14) | 0.30, [0.02, 0.54] (0.04) | 0.19, [−0.10, 0.45] (0.20) |
| Unusual thought content | 0.07, [−0.22, 0.35] (0.65) | 0.24, [−0.05, 0.49] (0.10) | 0.12, [−0.17, 0.40] (0.40) | 0.35, [0.07, 0.58] (0.02) | 0.04, [−0.25, 0.32] (0.81) | 0.15, [−0.14, 0.42] (0.30) |
| Disorientation | 0.17, [−0.12, 0.43] (0.26) | 0.03, [−0.26, 0.32] (0.83) | 0.37, [0.09, 0.59] (0.01) | 0.37, [0.09, 0.59] (0.01) | 0.06, [−0.23, 0.34] (0.70) | 0.16, [−0.14, 0.43] (0.29) |
| Poor attention | 0.24, [−0.05, 0.50] (0.10) | 0.11, [−0.18, 0.38] (0.47) | 0.33, [0.04, 0.56] (0.02) | 0.35, [0.07, 0.58] (0.01) | 0.11, [−0.18, 0.38] (0.47) | 0.27, [−0.02, 0.51] (0.07) |
| Poor insight | 0.14, [−0.15, 0.41] (0.33) | 0.15, [−0.14, 0.42] (0.30) | 0.08, [−0.21, 0.36] (0.61) | 0.22, [−0.07, 0.48] (0.13) | 0.18, [−0.11, 0.45] (0.22) | 0.30, [0.01, 0.54] (0.04) |
| Disturbance volition | 0.48, [0.22, 0.67] (<.01) | 0.26, [−0.03, 0.51] (0.08) | 0.41, [0.14, 0.63] (<.01) | 0.44, [0.17, 0.64] (<.01) | 0.31, [0.02, 0.54] (0.04) | 0.10, [−0.19, 0.38] (0.48) |
| Poor impulse control | −0.14, [−0.41, 0.15] (0.34) | −0.01, [−0.30, 0.28] (0.95) | −0.02, [−0.31, 0.27] (0.88) | −0.08, [−0.36, 0.21] (0.58) | −0.17, [−0.44, 0.12] (0.25) | 0.04, [−0.25, 0.32] (0.78) |
| Preoccupation | 0.28, [−0.01, 0.52] (0.06) | 0.42, [0.15, 0.63] (<.01) | 0.34, [0.05, 0.57] (0.02) | 0.63, [0.42, 0.78] (<.01) | 0.34, [0.06, 0.57] (0.02) | 0.13, [−0.16, 0.41] (0.37) |
| Social avoidance | 0.49, [0.24, 0.68] (<.01) | 0.57, [0.33, 0.73] (<.01) | 0.45, [0.19, 0.65] (<.01) | 0.45, [0.19, 0.65] (<.01) | 0.29, [0.00, 0.53] (0.05) | 0.25, [−0.04, 0.50] (0.09) |
| CDSS | 0.11, [−0.19, 0.38] (0.48) | 0.15, [−0.14, 0.42] (0.32) | 0.06, [−0.23, 0.34] (0.71) | 0.20, [−0.09, 0.46] (0.18) | 0.16, [−0.14, 0.43] (0.29) | −0.04, [−0.32, 0.25] (0.81) |
| AIMS | 0.01, [−0.28, 0.30] (0.93) | 0.22, [−0.08, 0.47] (0.14) | 0.10, [−0.19, 0.38] (0.50) | 0.29, [0.00, 0.53] (0.05) | −0.07, [−0.35, 0.22] (0.62) | 0.03, [−0.26, 0.31] (0.84) |

Abbreviations: Spearman's $\rho$ values with 95% confidence intervals and raw *p*-values. CDSS = Calgary Depression Scale for Schizophrenia; AIMS = Abnormal Involuntary Movement Scale.

group and the PANSS-PNS group captured two partially overlapping but distinct subsets of patients. While the BNSS-MS group was characterized by more severe negative symptoms overall, the PANSS-PNS group was defined by relatively lower contamination from secondary symptoms.

To our knowledge, this is the first validation of the BNSS in a Romanian-speaking clinical population and one of the few conducted in samples that include individuals in the acute phase of schizophrenia (Gehr et al., 2019a). Beyond broader considerations of generalizability, the use of an inpatient sample is additionally supported by the national representativeness of this setting, given that health resources in Romania remain disproportionately concentrated in hospital-based care rather than outpatient services (Radu et al., 2021; Păun et al., 2023). Regarding the cultural composition of the sample, the study population is reflective of national demographics, as it was drawn from one of the country's major urban centers. Although immigration has not yet significantly impacted the psychiatric landscape of the region, cultural diversity is evident in the local ethnic mix. This included several participants whose first language was not Romanian (primarily ethnic

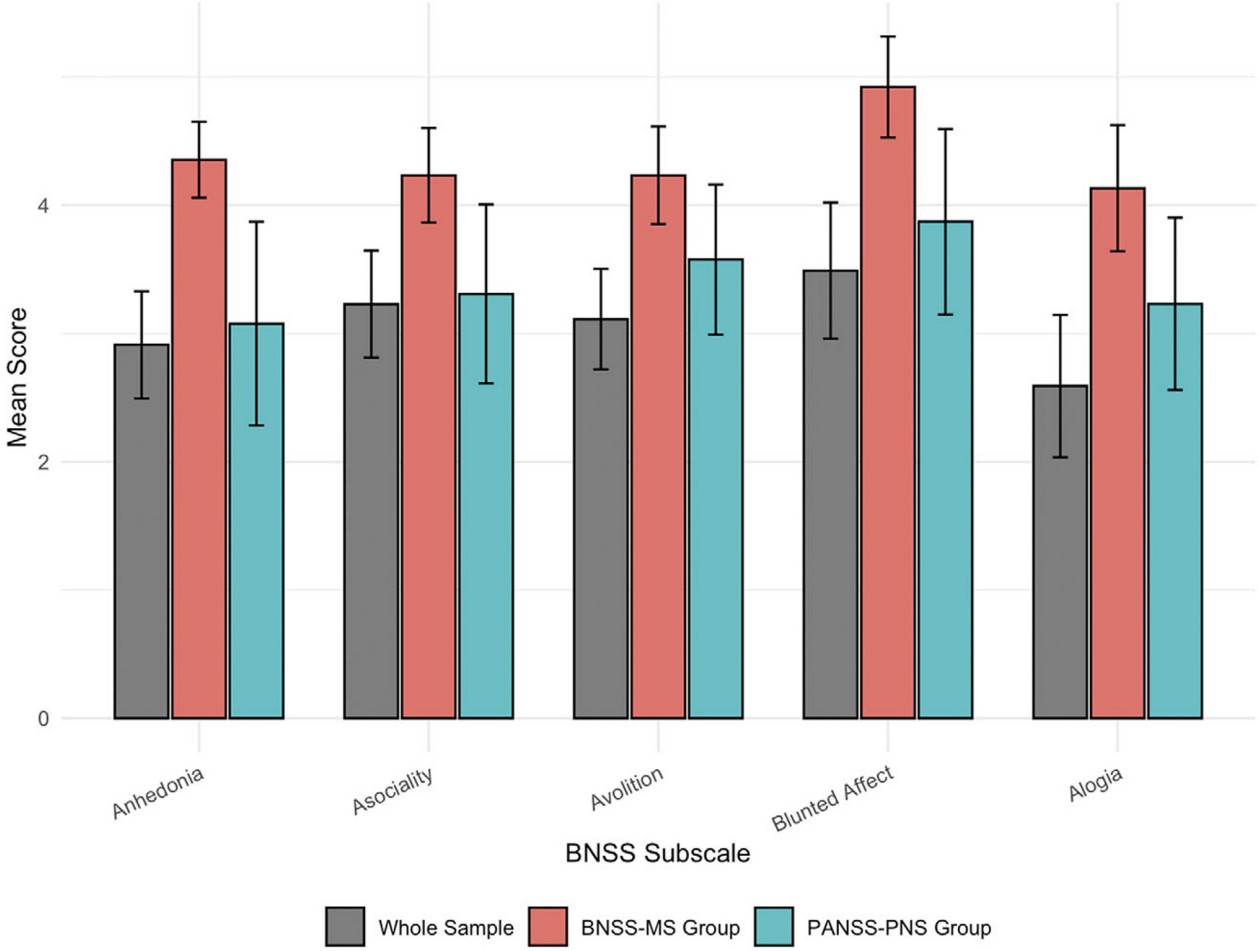

**Figure 1.** BNSS subscale mean scores: Whole sample vs. BNSS-MS & PANSS-PNS groups. Bar plots display mean BNSS subscale scores for the whole sample (gray), the BNSS-defined group (red) and the PANSS-defined group (blue), based on Friedman tests with *post hoc* comparisons. Blunted affect was significantly higher than alogia only in the whole sample ($p < .01$) and in the BNSS-defined group ($p = .01$). Groups include participants meeting exclusive or overlapping criteria. Error bars represent 95% confidence intervals around the means.

Hungarians), for whom informal language screening was conducted during the pre-interview phase to ensure adequate comprehension.

Construct validity analyses supported both a two-factor structure – MAP and EXP – that explained 64% of the variance, as well as the theorized five-factor model, which yielded the best fit in CFA. These results are broadly consistent with the original validation study (Kirkpatrick et al., 2011), which reported 71% variance explained by a two-factor model, and with recent cross-cultural findings supporting multidimensional representations of negative symptoms (Gehr et al., 2019b; Tatsumi et al., 2020). Although parallel analysis initially suggested a single dominant factor – likely due to the limited sample size – both theoretical and empirical considerations justified exploring a more nuanced factorial structure. Importantly, the lack of normal distress item showed low loadings in exploratory analysis and was excluded from the CFA and validity analyses, consistent with recent reviews advising against its inclusion in core negative symptom assessments (Weigel et al., 2023). Convergent validity was demonstrated by a very strong correlation between BNSS total and PANSS negative scores ($\rho = .90$, $p < .001$), and a moderate correlation with PANSS total ($\rho = .55$, $p < .001$), indicating that BNSS captures the core negative features assessed by standard instruments. Discriminant validity was supported by nonsignificant correlations with PANSS positive ($\rho = -.12$), CDSS ($\rho = .15$) and AIMS ($\rho = .12$) scores. Notably, the correlation with PANSS general was moderate but significant ($\rho = .42$, $p = .02$), reflecting some shared variance in overlapping constructs such as volition and attention. The BNSS–PANSS negative link remained virtually unchanged when controlling for CDSS and AIMS scores (partial $\rho = .92$, $p < .001$), further supporting the scale's robustness. Only one other BNSS validation study explicitly included acutely ill patients, conducted by Gehr *et al.*, who validated the Danish version. Interestingly, while our results showed no significant association between BNSS total scores and PANSS positive symptoms ($\rho = -.12$, $p = .42$), Gehr *et al.* reported moderate and statistically significant correlations between the BNSS and PANSS positive subscale, suggesting a greater degree of symptom overlap in their sample (Gehr et al., 2019a). Internal consistency was excellent (Cronbach's $\alpha = .94$ for the total scale, .80–.93 across subscales), in line with the original and cross-cultural studies (Kirkpatrick et al., 2011; Strauss et al., 2012). Inter-rater reliability, tested in a subsample of 10 double-rated patients, yielded an ICC(A,1) = .98, 95% CI (.42, 1.00), consistent with prior validations of BNSS (Mané et al., 2014; Mucci et al., 2015). While the wide confidence interval reflects the small sample size, the point estimate suggests excellent agreement between raters.

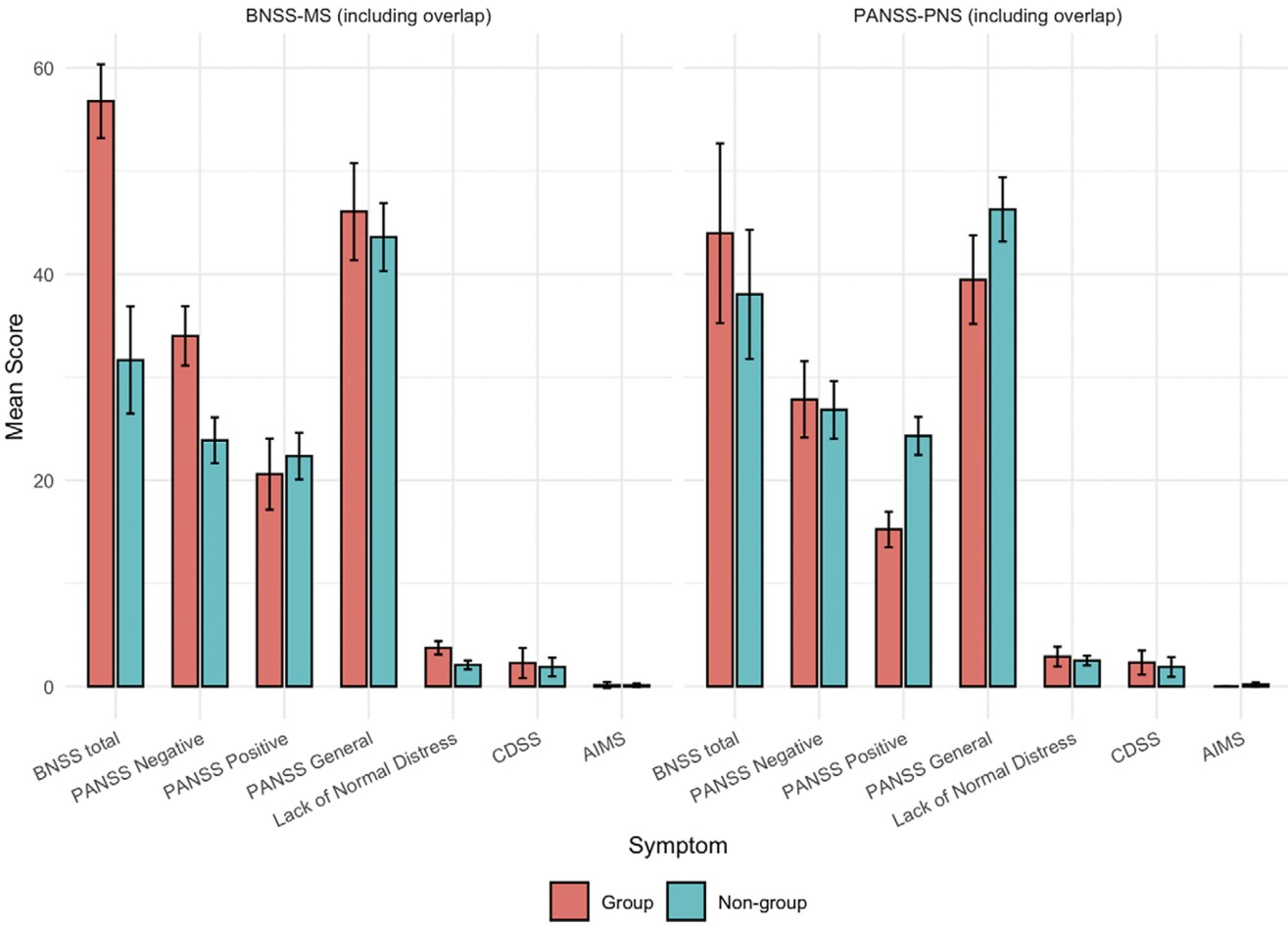

**Figure 2.** Negative-symptom severity and contamination scores including overlapping cases by BNSS and PANSS group definitions. "Group" refers to participants meeting the negative symptom criteria within each classification panel; "Non-group" includes all other participants. BNSS total and PANSS negative reflect negative symptom severity, while AIMS, CDSS, PANSS positive and general symptoms index potential secondary (contaminating) symptoms. Three sets of comparisons (t-test or Wilcoxon, as appropriate) were conducted: (1) BNSS-MS *vs.* PANSS-PNS groups, (2) group *vs.* non-group within BNSS-based classification and (3) within PANSS-based classification. BNSS total and PANSS negative scores were significantly higher in the BNSS-MS group compared to the PANSS-PNS group and within the BNSS-defined grouping (both $p < .01$). No significant differences in negative symptoms were observed within the PANSS-defined grouping. Error bars indicate 95% confidence intervals around the means.

In our sample, blunted affect emerged as the most prominent symptom domain, showing the highest mean scores across BNSS subscales and significantly exceeding Alogia in severity. This finding, although not commonly emphasized in previous validation studies, may reflect the acute clinical status of some participants, where psychotic and general psychopathology symptoms could obscure other negative domains. Indeed, when analyses were restricted to participants meeting criteria for severe negative symptoms and fewer contamination symptoms (PANSS-PNS), blunted affect no longer stood out significantly from the other subscales, suggesting that its initial prominence may be, at least in part, state dependent. Blunted affect also strongly correlated with PANSS negative and was moderately associated with AIMS scores ($\rho = .29$, $p = .05$), suggesting potential measurement contamination due to extrapyramidal symptoms. This warrants further investigation, particularly in heterogeneous or acutely symptomatic samples. The lack of normal distress item demonstrated a moderate and statistically significant correlation with the PANSS negative subscale. Within the general psychopathology subscale, it showed inverse associations with individual affective items – somatic concern, anxiety, tension, and depression – as well as with the CDSS total score. A weak positive correlation was also observed with the

poor insight item. These patterns are consistent with the conceptualization of the lack of normal distress item as a noncore negative symptom that nonetheless captures a clinically relevant dimension of reduced emotional insight, which may co-occur with negative symptoms across varying levels of severity, as shown in other previous studies (Gehr et al., 2019a).

Sociodemographic variables were largely unrelated to BNSS subscales, except avolition, which was significantly elevated in unemployed patients ($p = .03$). Our definition of unemployment included both those without jobs and those on psychiatric disability, and thus indirectly captured the impact of illness on global functioning. This finding is consistent with previous work showing strong links between negative symptoms, especially avolition and poor psychosocial outcomes (Rabinowitz et al., 2013; Bischof et al., 2016; Mucci et al., 2019).

To further characterize clinical phenotypes, we applied two grouping strategies: BNSS-MS and PANSS-PNS. The two classifications showed only slight agreement (Cohen's $\kappa = .19$), with BNSS-MS identifying a larger subset of patients with clinically relevant negative symptoms, as well as significantly higher negative symptom severity (BNSS total and PANSS negative scores) than the PNS group. These results are consistent with previous evidence (Mucci

et al., 2019) that BNSS is more sensitive than PANSS in detecting negative symptoms. Notably, BNSS-MS participants also showed elevated PANSS positive scores relative to the PNS group. While this could indicate symptom contamination, it likely reflects the generally higher PANSS total in our sample, which included individuals with moderate psychotic activity. Importantly, only the BNSS-MS grouping yielded significant differences in lack of normal distress, with higher scores among group members, supporting prior proposals (Kirkpatrick et al., 2011) that reduced distress may help isolate primary negative symptoms. These findings suggest that while BNSS-based classification may offer greater clinical sensitivity to core negative symptoms, it may also lack specificity, particularly in samples that include acutely psychotic patients, as highlighted in previous research (Gehr et al., 2019a). Finally, the proportion of BNSS-MS cases in our sample (31.91%) is consistent with estimates of the deficit syndrome subtype in schizophrenia, typically 20–30% in clinical samples (Buchanan, 2007), further supporting the clinical validity of the BNSS as a tool for subgroup identification.

Several limitations of the current study should be acknowledged. First, the relatively small sample size represents a significant limitation; future multicenter studies with larger sample sizes are necessary to validate the BNSS within Romanian-speaking populations further. Second, the diagnostic inclusion criterion was restrictive, limited only to schizophrenia; other psychiatric diagnoses that involve significant negative symptoms, such as major depressive disorder and bipolar disorder, should be included in future BNSS validation studies to broaden generalizability. Third, inter-rater reliability was assessed in only a subset of participants (10 out of 47 subjects), due to the need to adapt to varying hospitalization durations and the clinical course of the patients. Specifically, inter-rater assessments were conducted with patients who had longer hospital stays and a more predictable clinical evolution, allowing for the feasible scheduling of double ratings. Subsequent research should aim to double-rate a larger proportion – or ideally the entire sample – to strengthen the reliability data. Fourth, our discriminant validity assessment was limited by the omission of cognitive symptom evaluation, which is an important source of potential pseudospecificity in negative symptom measurement. Future studies should address this limitation by incorporating cognitive assessment tools, such as the Brief Assessment of Cognition in Schizophrenia (BACS) or more clinically practical instruments like the Montreal Cognitive Assessment (MoCA) to better delineate between cognitive and negative symptoms. Finally, the study did not directly include an assessment of the associations between the five BNSS-measured negative symptoms and psychosocial functioning, even though negative symptoms are recognized as key predictors of poor functional outcomes in schizophrenia. Specific evaluation of the relationship between the deficit phenotype and occupational functioning using dedicated scales, such as the Social and Occupational Functioning Scale (SOFS), was beyond the scope of the present study. Our objective was limited to screening for associations between negative symptoms and demographic variables, such as social status. Future research should aim to explicitly investigate these associations using standardized measures.

## Conclusions

Four main conclusions emerged from the current study: (1) The BNSS (Romanian: "Scala Scurtă a Simptomelor Negative") demonstrated strong psychometric properties within a Romanian-speaking clinical population, confirming its suitability for reliably assessing negative symptoms across heterogeneous schizophrenia samples, including acutely ill patients. (2) Blunted affect emerged as a particularly prominent negative symptom, visibly distinct and prominent even amidst acute psychotic presentations. (3) Avolition appeared to be the negative symptom most strongly associated with social disability, underscoring its relevance in predicting functional impairment. (4) The BNSS exhibited superior sensitivity compared to PANSS in identifying clinically meaningful severe negative symptoms, even when applying minimal severity threshold criteria in complex clinical presentations. This advantage highlights its value as a robust screening instrument for detecting deficit phenotypes in schizophrenia.

**Open peer review.** To view the open peer review materials for this article, please visit http://doi.org/10.1017/gmh.2025.10037.

**Acknowledgments.** The authors gratefully acknowledge the support of the original BNSS authors, particularly Dr. Kazunori Tatsumi, Head of the Translation Department, for providing the necessary materials and guidance for the application of the scale in Romanian.

**Author contribution.** CIM was the lead contributor to the conception and design of the study, as well as to the acquisition, analysis and interpretation of data. DCG contributed to participant recruitment and symptom rating. OOC contributed to the conceptual development of the study. CAC and MFS assisted in identifying appropriate participants. IVM provided the final evaluation and approved the manuscript for submission.

**Financial support.** This work was supported by the university grant for doctoral research (PCD 2021/2022).

**Competing interests.** The authors declare none.

**Ethics statement.** This study was approved by the Ethics Committee of Iuliu Hațieganu University of Medicine and Pharmacy, Cluj-Napoca (approval number: AVZ56/04.04.2023). All participants provided written informed consent before enrollment.

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
