## [Reviewer Report]

This study offers the first validation of the Brief Negative Symptom Scale (BNSS) in a Romanian-speaking population and one of the few conducted in a heterogeneous schizophrenia sample, including acutely ill patients. The background of this study emphasis the need to assess reliably the negative symptoms in schizophrenia, which are critical to functional outcomes in patients living with this diagnosis. The objectives of this research were: To validate the BNSS in a Romanian clinical sample, explore its psychometric properties, and compare BNSS-based and PANSS-based

classifications of severe negative symptoms. The results indicate that the BNSS has notable psychometric properties, making it a highly reliable and useful instrument compared to previous versions that do not include all five factors. In addition, the five-factor structure was confirmed by a detailled factor analysis. BNSS total scores correlated strongly with PANSS Negative, but not with positive, depressive, or motor symptoms. Blunted Affect emerged as the most prominent subscale. The BNSS-MS group captured more severe cases than PANSS-PNS and showed greater symptom

burden and higher distress scores. In conclusion, the research demonstrates that the Romanian version of BNSS is valid and sensitive for detecting negative symptoms, outperforming PANSS in identifying clinically significant subgroups.

This study has several positive aspects: a) This is a quantitative study with several phases that demonstrates the psychometric value of the instrument, properly indicating the procedures; b) In methodological terms, it is at the forefront of instruments that measure negative symptoms in schizophrenia, as well as to adequately discern between different severities of this diagnosis; c) For these reasons, it is a high-impact investigation in the psychometrics of the instrument as well as to validate its applied use in different clinical devices; d) It is written clearly, precisely, and adequately meets quantitative research standards.

Although this research is noteworthy in all the aspects already discussed, I believe there are some minor improvements that could be made. These are suggestions rather than necessary changes, but given its relevance and impact, I believe it would be worthwhile to refine this valuable research. Below I present some extracts from the writing and the number of lines in which it is found, and the recommendation that I would suggest to perfect the study.

“a diagnosis of schizophrenia according to the Diagnostic and Statistical Manual of Mental Disorders, Fifth Edition (DSM-5).” (lines 131-132)

Why wasn’t the most up-to-date version of the DSM-5-TR used? Beyond having to use it, I ask so that the reasons are explicit in the document (whether for practical, technical, or service availability reasons, among others), considering that the general recommendation is to use the most up-to-date version for diagnosis. Since it doesn’t have major effects on the diagnosis of schizophrenia, it doesn’t significantly affect it, but I think it would be prudent to establish what conditions lead to using the DSM-5.

In the construction of the justification of the research problem, I would incorporate some data regarding the situation in health services and data/figures on the national reality regarding mental health, schizophrenia and how interventions have been carried out in your country, in order to briefly add a social and/or practical relevance to the study, rather than just the methodological dimension.

"A multidisciplinary panel reviewed all versions, and minor adjustments were made to

ensure clarity, cultural relevance, and clinical suitability." (lines 168-169)

¿What minor adjustmentes? I would recommend explaining what changes were made, even if they are minor, or just listing a few to understand how the final adapted version turned out, and whether it relates to word (semantic) or grammatical understanding. This is because certain words could have different meanings depending on the cultural context.

“Normality was assessed using the Shapiro–Wilk test to inform the use of non-parametric statistics.” (lines 176-177)

I would suggest to include the Shapiro-Wilk test results in Table 1 or the resulting value in the same text, in order to empirically reinforce the decision of non-parametric tests, even though it may be predictable considering the sample size.

“Fourthly, our discriminant validity assessment was limited by the omission of cognitive symptom evaluation, which is an important source of potential pseudospecificity in negative symptom measurement.” (lines 363-365)

I would recommend exploring a possible way to address this aspect of cognitive symptoms in future research, considering the significant results obtained with this scale, to enhance it especially with regard to attention deficits.

To the authors of the article, my congratulations for this work. I offer these suggestions primarily to further enhance your work, but only consider them if they make sense to you as authors.

---

## [Reviewer Report]

This is a highly relevant and timely study that makes an important contribution to the field of cross-cultural mental health assessment. The following are minor suggestions aimed at enhancing the clarity and rigour of the manuscript. The work is already strong, and these comments are intended to support your efforts in further strengthening it.

<b>Impact Statement:</b>

• Line 6: I recommend using a word other than “superior,” as it may carry unintended connotations.

<b>Materials and Methods:</b>

*Participants:*

• Since this study involved inpatients with schizophrenia, it is important to describe how participants' capacity to provide informed consent was assessed. For instance, what procedures were followed if a participant was experiencing a psychotic episode?

• Regarding the exclusion criterion of insufficient fluency in Romanian, it would be relevant to explain how language proficiency was assessed. For example, did you include only participants whose first language was Romanian, or also those for whom Romanian was a second language? This should also be addressed in the Discussion section.

• Are there any additional relevant participant characteristics that could be reported, particularly given the cultural adaptation of the scale beyond translation? For instance, ethnic backgrounds, migration status, etc.

*Translation and Cultural Adaptation:*

• Could you provide more details about the multidisciplinary team mentioned in line 168? For example, what were their professional backgrounds? Did the team include individuals with lived experience of schizophrenia?

*Procedure:*

• You mentioned that inter-rater reliability was assessed using a subsample, which is later discussed as a limitation. Could you elaborate on the rationale for selecting a subsample of 10 participants rather than conducting the analysis with the full sample? Since you list this as a limitation, it would be relevant to clarify this decision.

*Ethical Considerations:*

• Did you follow any specific ethical protocols for working with participants with schizophrenia? For example, were there procedures in place for identifying participants who were not in a condition to participate or who required additional support? Were any potential risks to participants assessed?

<b>Results:</b>

• It would be helpful to report confidence intervals throughout the results section (not only for the intraclass correlation coefficient), as relying solely on p-values can be limiting. Confidence intervals offer more meaningful insight.

• Line 218: You mention that inter-rater reliability was assessed in a subsample of 12 participants, but other sections refer to 10. Please clarify and ensure consistency.

• Table 2: Add a footnote stating explicitly that inter-rater reliability was assessed in a subsample, including the number of participants.

• Ensure consistency in decimal reporting throughout the results (e.g. use the same number of decimal places for all statistics).

• Figures 1 and 2: Can you include confidence intervals as error bars? This would enhance the visual comparison of findings.

<b>Discussion:</b>

The limitations section could be improved as follows:

• For the second limitation, could you provide examples of the “other psychiatric diagnoses” mentioned in line 359?

• For the third limitation, please refer to the earlier comment about the subsample used for inter-rater reliability.

• For the fifth limitation, could you elaborate on why the mentioned analysis was not conducted?

---

## [Editor Report]

Thank you for submitting your manuscript. Both reviewers found your work to be a significant contribution to the field, with strong methodological rigor and clear, precise reporting. The suggestions concern minor revisions to enhance clarity and transparency—for example, elaborating on informed consent procedures, describing the multidisciplinary team in more detail, clarifying methodological choices, and providing additional justification for diagnostic criteria. We encourage you to address these points, as they will further strengthen your already valuable study, and we look forward to receiving your revised manuscript.